# The Effects of Workplace Stressors on Dietary Patterns among Workers at a Private Hospital in Recôncavo of Bahia, Brazil: A Longitudinal Study before and during the COVID-19 Pandemic

**DOI:** 10.3390/ijerph20054606

**Published:** 2023-03-05

**Authors:** Lorene Gonçalves Coelho, Priscila Ribas de Farias Costa, Luana de Oliveira Leite, Karin Eleonora Sávio de Oliveira, Rita de Cássia Coelho de Almeida Akutsu

**Affiliations:** 1Health Science Centre, Federal University of Recôncavo of Bahia, Santo Antônio de Jesus, Bahia 44574-490, Brazil; 2School of Nutrition, Federal University of Bahia, Salvador, Bahia 40110-150, Brazil; 3Department of Nutrition, Campus Darcy Ribeiro, University of Brasilia, Asa Norte, Distrito Federal, Brasília 70910-900, Brazil

**Keywords:** occupational stress, demand–control model, workplace stressors, dietary patterns, COVID-19 pandemic

## Abstract

Working in a hospital environment is known for presenting unhealthy features that affect the workers’ health—features which have currently been intensified due to the COVID-19 pandemic. Hence, this longitudinal study aimed to ascertain the level of job stress before and during the COVID-19 pandemic, how this changed, and its association with the dietary patterns of hospital workers. Data on sociodemographic, occupational, lifestyle, health, anthropometric, dietetic, and occupational stress were collected before and during the pandemic from 218 workers at a private hospital in the Recôncavo of Bahia, Brazil. McNemar’s chi-square test was used for comparison purposes, Exploratory Factor Analysis to identify dietary patterns, and Generalized Estimating Equations to evaluate the interested associations. During the pandemic, participants reported increased occupational stress, shift work, and weekly workloads, compared with before the pandemic. Additionally, three dietary patterns were identified before and during the pandemic. No association was observed between changes in occupational stress and dietary pattens. However, COVID-19 infection was related to changes in pattern A (0.647, IC95%0.044;1.241, *p* = 0.036) and the amount of shift work related to changes in pattern B, (0.612, IC95%0.016;1.207, *p* = 0.044). These findings support calls to strengthen labour policies to ensure adequate working conditions for hospital workers in the pandemic context.

## 1. Introduction

Work is an essential part of human life, since it provides their means of subsistence, dignifying them as beings who live in society. At the same time, it can submit individuals to work environments that are harmful to their health [1]. Unsafe work environments can predict physical and mental strain and occupational stress, as evidenced by Tao et al. [2] in their study with geological investigators, and by Alrawad et al. [3] in their work with mineworkers.

Working in a hospital environment is also known to present unhealthy characteristics which affect workers’ health, and this has intensified due to the COVID-19 pandemic. As a result, studies reporting an increase in psychological distress and the occupational stress levels of these individuals have been recurrent in the literature [4,5].

Before the COVID-19 pandemic, Ribeiro et al. [6] found that 27.4% of workers of a hospital in southern Brazil were exposed to intermediate and high levels of work stress, and Fang et al. [7] identified that 20.5% of nurses in a university hospital in southern Taiwan were exposed to high psychological demands in the work process. During the pandemic, Saind and El-Shafei [8] found that 75.2% of nurses in a COVID-19 triage hospital in Sharkia Governorate, Egypt, had a high level of occupational stress, and Magnavita et al. [9] reported that 71.1% of physicians in a hub hospital in Latium, Italy, were also submitted to high levels of stress at work.

The increase in stress levels among hospital workers during the pandemic can be related to the emergence of situations that they had rarely experienced before, such as increased stress in patient care, a feeling of high risk in work performance, concern for their health, with the health of family members, and with self-isolation [10]. Furthermore, with the increased number of hospitalizations due to COVID-19, there were changes in the structure and organization of work in hospitals, imposing an even more harmful work environment on workers [5].

In turn, it is known that occupational stress, i.e., high psychological demands at work, is associated with changes in workers’ lifestyles and health [11,12,13,14], especially with regards to food. Coelho et al. [10], in an integrative review, demonstrated the occurrence of negative changes in nurses’ eating habits because of work. Nuhu et al. [12] also found changes in the food consumption of nurses from two hospitals in Ghana; those with high levels of stress at work had low caloric intake. Additionally, Islam et al. [14] found that health professionals at a field hospital against COVID-19 in Gazipur, Bangladesh, had their meals irregularly.

Thus, it is essential to consider changes in the behaviour and food consumption of workers in stressful situations due to the well-known association between food and nutrition and non-communicable chronic diseases, making them important factors to maintain and promote good health [12,13,14]. In addition, for workers, when food consumption is inadequate, there may still be a reduction in work capacity and productivity, which makes the process a perverse cycle where the reduction in productive capacity compromises income and, consequently, the ability to provide good food [1].

With regards to ways of evaluating food consumption, this has been performed while only considering the isolated consumption of nutrients or foods for several years. However, food and their nutrients are consumed together, and interact with each other, having been established in the literature that their real effects can only be observed when the entire eating habit is considered [12,15].

Therefore, the number of studies using the assessment of dietary patterns has increased in epidemiological studies, since foods are analysed synergistically and simultaneously in this approach, considering complex combinations between nutrients, which facilitates the description and knowledge of the effect of food on the health and disease process [13,14,15].

There is a lack of studies specifically on occupational stress and dietary patterns, which seek to understand this relationship within a theoretical framework, reinforcing the need for further investigations, for a better understanding of the phenomenon of eating at the interface of occupational stress, and over time, limiting themselves to evidences from cross-sectional studies that do not allow understanding causality relationships, or that deal with food as secondary feeding or the isolated consumption of nutrients. Thus, the need to deepen investigations is reinforced, adopting a robust study design, such as a prospective cohort, for better understanding the phenomenon of food in the interface of occupational stress, which justifies the accomplishment of the present work. Accordingly, we hypothesized that high levels of occupational stress contribute to changes in the dietary patterns of hospital workers.

Thus, the objective of this study is to verify the level of occupational stress before and during the COVID-19 pandemic, and its change and association with the dietary pattern of workers at a hospital in the Recôncavo of Bahia, Brazil.

## 2. Materials and Methods

### 2.1. Study Design and Sample

This is a longitudinal study that used baseline and the first follow-up data from one of the hospitals in the cohort “Evaluation of Food and Nutrition Services in three hospitals in the health network of Salvador, Bahia”. Only one of the study hospitals was included in this study, as the other sites withdrew consent to participate during the COVID-19 pandemic. The hospital in question is in the town of Santo Antônio de Jesus, Bahia, and had a staff of 371 workers in 2019. Initially, all 371 workers were invited to participate in the study; however, according to the inclusion and exclusion criteria described below, as well as the losses that occurred during the study, the final sample included 218 workers from different sectors of the hospital, as described in other publications [16,17].

### 2.2. Eligibility Criteria

Workers of both sexes, aged over 18 (Brazilian majority), who agreed to participate in the research by signing a free and informed consent form were eligible. Individuals with problems that compromised taking anthropometric measurements were not included: those who had recent abdominal surgeries and suffer from abdominal lesions, tumours, hepatomegaly, splenomegaly, ascites, and amputees; as well as pregnant women, or those who had given birth in the last six months, due to changes in body composition characteristics at these stages of life [18].

### 2.3. Data Collection

Data collection was performed by a team of nutritionists trained in research protocol. Sociodemographic, occupational, lifestyle, health, anthropometric, and occupational stress variables were collected between May and October 2019 (before the pandemic—baseline), and between October and November 2020 (during the pandemic—first follow-up), considering the same instruments, techniques, and procedures in both evaluation periods.

#### 2.3.1. Sociodemographic, Occupational, Lifestyle, and Health Variables

The variables in question were collected through a structured questionnaire. Gender, age, skin colour/ethnicity [self-reported], marital status, education, and income were the sociodemographic variables. Occupational variables included occupation [health professional, or other], how long they had worked at the hospital [months], weekly workload, and shift work. With regards to lifestyle, the variables of smoking and alcohol consumption habits, and level of physical activity were evaluated through the reduced and validated version of the International Physical Activity Questionnaire, with workers classified as having low (<600 metabolic equivalents (MET)—minutes/week), moderate (600 to 3000 MET-minutes/week), and high levels of physical activity (≥3000 MET-minutes/week) [19]. In relation to health, the variables of family history for non-communicable chronic diseases, perception of one’s own health, and self-reported contamination/infection by COVID-19 were considered.

#### 2.3.2. Anthropometric Variables

Weight, height, and waist circumference (WC) formed the anthropometric variables. Weight was measured using a portable digital scale with bioimpedance on a platform (Full Body Sensor—Body Composition Monitor and Scale, model HBF-516, OMRON^®^ brand). Respondents were weighed following techniques described in the literature [20]. Height was measured using a portable stadiometer (Alturaexata^®^). The technique used is recommended by the World Health Organization (WHO) [20]. The Body Mass Index (BMI) was calculated from weight and height measurements, represented by the Kg/m^2^ ratio [20]. The cut-off point used to classify the nutritional status of workers, according to the BMI, was that proposed by the WHO [21]. The WC was measured using a flexible, inelastic measuring tape, following WHO recommendations [20]. This measure was used to predict the risk of metabolic and cardiovascular complications in workers, while considering the cut-off points proposed by the WHO [22].

#### 2.3.3. Dietary Variables

The Brazilian Longitudinal Study of Adult Health—(ELSA-Brasil) food frequency questionnaire (FFQ) was used to obtain an estimate of usual food consumption in the twelve months prior to the study evaluation periods [23].

The ELSA-Brasil FFQ presents a list of foods made up of 114 items and is structured in three sections: (1) food/preparation, (2) measures of consumption portions, and (3) consumption frequency, with eight categories: more than 3 times/day, 2–3 times/day, once/day, 5–6 times/week, 2–4 times/week, once/week, 1–3 times/month, and never/almost never [23].

We also highlight that the 114 items on the FFQ are categorized into the following food groups: “breads, cereals, and tubers”, “fruits”, “vegetables and legumes”, “eggs, meat, milk, and dairy products”, “pasta and other preparations”, “candy”, and “beverages” [23]. However, for the purposes of the analysis of this study, the foods in the “breads, cereals, and tubers”, “vegetables and legumes”, “eggs, meat, milk, and dairy products”, and “beverages” groups were reorganized into “breads and cereals” and “tubers”, “vegetables”, “legumes”, “oilseeds”, “eggs”, “meat”, “milk and dairy products”, “fats”, “beverages”, and “sugary drinks”, respectively, according to Food-Based Dietary Guidelines for the Brazilian Population [24].

In addition, data from FFQ consumption frequencies (daily, weekly, and monthly) were converted into daily consumption portions, to use a time unit in the analyses, as proposed by Coelho [25].

#### 2.3.4. Occupational Stress Variables

The instrument used to assess occupational stress was the JCQ in its reduced version, translated and validated for the Brazilian population. The JCQ consists of 17 questions, divided into the following dimensions: (1) demand, (2) control, and (3) social support, with the response options presented on a Likert scale (1–4) [26].

The “demand” dimension comprises five questions that address pace, workload, time, conflicting demands, and work effort. There are six questions for the “control” dimension, related to learning, skill, creativity, repetitiveness, responsibility, and decision-making. The “social support” dimension has six questions about interpersonal relationships [26].

To classify occupational stress, we used the Demand–Control Model, which makes the theoretical assumption that the coexistence of great psychological demands and low control in the work process generate job strain, which results in increased stress at work [26]. Following this, participants were classified as having “high occupational stress” if they report above the median score in the “demand” dimension and below the median score in the “control” dimension of the JCQ, and “low occupational stress” otherwise [27].

### 2.4. Identification of Dietary Patterns

Identification of workers’ dietary patterns was carried out through factor analysis of the principal components at both points in time of the study (before and during the COVID-19 pandemic) and considering the 14 food groups described above.

This type of analysis reduces by one factor (dietary pattern) the food groups that are correlated with each other, but that are independent and do not contribute to other patterns in the analysis, indicating the factor loading of the correlation between the food group and its respective factor [28].

To verify the applicability of the data to the factor analysis, the Kaiser–Meyer–Olkin (KMO) test and the Bartlett sphericity test were used, considering acceptable values above 0.60 and *p* < 0.05, respectively. The KMO assesses the factor model adequacy through partial correlations and their respective weights, as the closer to 1, the higher the factor model adjustment, and values lower than 0.60 are not accepted [29,30]. Bartlett’s sphericity test considers there are no correlations between the data, i.e., the correlation matrix generated in the analysis is an identity matrix. Therefore, the factor model is suitable when it produces a correlation matrix that differs from the identity matrix, which is indicated by *p*-values ≤ 0.05 [29,30].

After verifying the adequacy of the factor analysis for the data set, each food group had its commonality assessed. The commonality reflects the level of connection between the variable (group) and the factor (pattern). It can vary from 0 to 1; the closer to 1, the higher the connection between them. In this study, variables with commonality values > 0.30 were considered as representative of the factor [29,30].

As for the number of factors selection, the criterion of eigenvalues or Kaiser’s criterion was used. As the eigenvalue is influenced by the total number of components of the analysis, generally high in studies on food consumption, some authors use a criterion of eigenvalues greater than 1 as a cut-off point to determine the total number of factors to be retained in the analysis [31]. Thus, this cut-off point was adopted since it allows better interpretability of dietary patterns and retains a smaller number of factors with the highest percentages of variance; being more representative of the food of the studied workers.

Finally, to improve the interpretability of food groups belonging to each dietary pattern and to obtain unrelated patterns, we use the Varimax orthogonal rotation method. At the end of the analysis, the rotated matrix was evaluated, and considering the sample size of this study, the variables that presented a factor loading > 0.30 in the retained factors characterized the dietary patterns [29,30].

The analyses were performed using STATA for MAC statistical software (Version 17.0, Stata Corp LP, College Station, TX, USA).

### 2.5. Identification of Variables

The dietary patterns identified through the factor analysis were named A, B, and C, and they were the outcome variables of this study. Their measurements were taken at the baseline, and after a minimum interval of twelve months’ follow-up, to assess changes over time. For integration into statistical models, the patterns were considered in their categorical form, adopting the 50th percentile (P50) (<P50 (0) and >P50 (1)) as the cut-off point.

Occupational stress, also assessed at the beginning and after a minimum twelve-month interval, was considered the main exposure in this study. Integration into statistical models took place in a categorical form: “absence or low levels of stress at work” (0) and “high levels of stress at work” (1). Similarly, other occupational characteristics considered stressors at work were considered as additional exposures: shift work, no (0) and yes (1), weekly workload, <44 (0) and >44 h, and infection by COVID-19, no (0) and yes (1).

The covariates of the study included: age (years), sex (women, men), educational level (<high school, >college), income (<3 minimum wages (MW), 3–5 MW, >5 MW), occupation (health professional, other), smoking status (current/ex-smoker, non-smoker), alcohol consumption (yes, no), physical activity level (low, medium, high), health self-perception (excellent/good, regular/bad), and nutritional status according to body mass index (underweight/normal range, overweight/obese) and waist circumference (low risk, increased/high risk).

### 2.6. Statistical Analyses

Descriptive statistical analysis expressed the categorical variables as absolute and relative frequencies, and the continuous variables as mean and standard deviation. Data normality was checked by the Shapiro–Wilk test. McNemar’s chi-square or Wilcoxon tests were used to compare the prevalence of occupational stress before and during the COVID-19 pandemic. Pearson’s chi-squared test and Student’s *t*-test were used to verify the distribution of dietary patterns according to the covariates of the study.

In order to assess the influence of occupational stress and additional exposures (shift work, weekly workload, and COVID-19 infection) on changes in dietary patterns A, B, and C before and during the pandemic, Generalized Estimating Equation models (GEE) were constructed. These are appropriate for categorical response variables and repeated measures, reflecting the relationship between outcomes and exposures, considering the correlation and interdependence between measures at each moment in time [32]. The GEE is able to produce more efficient and less biased estimates of correlated (repeated) data, since it considers the intra- and inter-individual correlation structure [32]. The matrix chosen for this study was the correlation matrix. 

Quasi-likelihood criterion (QIC), under the corrected independence model, was used to fit the models to the data, which is an adaptation of Akaike’s information criterion (AIC) method for GEE analyses. The QIC is calculated by comparing the quasi-likelihood of the independence model with the complete model. The lower the QIC, the better the model fits [33,34].

A model was built for each outcome variable (dietary patterns A, B, and C)—inserted into a categorical and time-variant form—depending on the main exposure variable (occupational stress) and additional exposures (shift work, working hours, weekly workload, and COVID-19 infection)—also categorized. Initially, univariate analysis was performed, and those with a *p* value lower than 20% were selected. These variables, together with those which showed potential for confounding in the bivariate analysis, were included in the model. Potential confounding variables were those associated with both exposure and outcome, expressed as a change of 10%, or more, in the association measure, compared with the reduced model measure [35]. Interaction terms were tested—built based on the literature and the data structure of the study—to assess the existence of modification of the effect of exposure variables on the outcome variable, using the maximum likelihood-ratio test (log likelihood-ratio test), evaluating the significance of the interaction term in the multivariate model. The variables which presented a significance of less than 5% remained in the final model.

The analyses were performed using STATA for MAC statistical software (Version 17.0, Stata Corp LP, College Station).

### 2.7. Ethical Aspects

This study was conducted according to the guidelines laid down in the Declaration of Helsinki and all procedures involving human subjects were approved by the School of Nutrition at the Federal University of Bahia Ethics Committee for ethical pertinence [36], under number 4,316,252. Written informed consent was obtained from all subjects.

In addition, in compliance with ethical assumptions, all workers who presented significant changes in the indicators evaluated were referred to local health services and remained in the study.

## 3. Results

At the baseline, the workers’ mean age was 32.60 (8.30). The average length of hospital work experience was 45.96 (35.72) months. In all, 41.70% of the workers were health professionals, while the remainder occupied other positions, such as administrator, cleaner, telephonist, and labourer, etc. With regards to educational level, 54.60% of the participants attended high school, and 45.40% subsequently took college or university courses. Most of the workers (52.30%) were married or had a common-law partner, and 42.20% were single. Further worker characteristics at the baseline are reported in Table 1.

With regards to the prevalence of work stress (primary exposure) among workers, and other occupational characteristics (additional exposures) before and during the COVID-19 pandemic, there was a 107% (14.20 versus 29.40%) increase in the high level of occupational stress, 26% (32.10 versus 39.40%) in the number of individuals working shifts, and 32% (22.90 versus 30.30%) in those working more than 44 h a week. All these differences were highly significant (McNemar’s chi-square test *p* < 0.001, *p* = 0.001 and *p* = 0.02, respectively).

In addition to the stress levels and occupational characteristics mentioned above, the contamination of workers by COVID-19 was also investigated, since it is considered a new stressor in the hospital work environment. In total, 67.40% (n = 147) of the individuals in the sample reported having tested positive for COVID-19 during the pandemic.

As for dietary patterns, three were identified in both assessment periods, i.e., before and during the COVID-19 pandemic. The three pre-pandemic patterns explained 45.51% of the sample variance and those during the pandemic explained 44.47%.

The first pattern identified before the pandemic (A) was characterized by the basic food groups for the Brazilian population (bread and cereals, and legumes), as well as pasta and other preparations, meat, milk and dairy products, fruits, candy, and fats, being responsible for 18.00% of the total variance. During the pandemic, the first pattern (A) was responsible for 17.20% of the variance. Tubers, eggs, vegetables, fruits, and oilseeds were the food groups with the positive, high factor loadings (>0.30) which characterized this pattern (Table 2).

The second pattern prior to the pandemic (B) was also characterized by the tubers, eggs, vegetables, fruits, and oilseeds groups. With regards to the second pattern during the pandemic (B), this was formed by bread and cereals, meat, fats, pasta and other preparations, candy, and sugary drinks groups (Table 2). The variance attributed to each of these patterns was 15.95% and 14.71%, respectively.

The third and final pattern (C) explained 11.56 and 12.56% of the total variance before and during the pandemic. In these two evaluation points in time, the characteristic groups were bread and cereals, legumes and beverages; and bread and cereals, milk and dairy products, fats and beverages, respectively (Table 2).

Distribution of the workers’ sociodemographic, lifestyle, and health characteristics, that is, the covariates of the study, according to dietary patterns A, B, and C are described in Table 3. Pattern A was associated with the workers’ income (*p* = 0.042), and pattern B to educational level (*p* = 0.009), while pattern C had both characteristics (*p* < 0.001 and *p* = 0.009, respectively).

With regards to the influence of occupational stress on changes in dietary patterns before and during the pandemic, no significant results were observed, according to the raw GEE models (Table 4). However, when considering additional exposures, a positive and significant association was confirmed between COVID-19 infection and changes in dietary pattern A (0.684, 95%CI0.079; 1.290, QIC 303677, *p* = 0.027), and between shift work and changes in pattern B (0.650, 95%CI0.053; 1.248, QIC 302894, *p* = 0.033) (Table 4).

These results were adjusted, considering the covariates of the study: the model between COVID-19 infection and changes in pattern A was adjusted for income, physical activity level, and nutritional status, according to the WC (0.631, 95%CI0.030; 1.231, QIC 302501, *p* = 0.031), with only the covariate nutritional status according to the WC (0.647, 95%CI0.044; 1.241, QIC 302952, *p* = 0.036) remaining as an adjustment in the final model. Adjustment of the model for shift work and changes in dietary pattern B was carried out through the covariates of educational level and income (0.611, 95%CI0.015; 1.206, QIC 30148, *p* = 0.044), with adjustment of the final model only including the educational level (0.612, 95%CI0.016; 1.207, QIC 300462, *p* = 0.044). In the four scenarios presented, statistical significance was maintained between exposures and outcomes.

## 4. Discussion

The results of this study revealed a significant increase in the high level of stress at work, as well as the number of individuals working shifts and more than 44 h per week during the COVID-19 pandemic. In addition, three dietary patterns were identified at both points in time of the study, with no association between these and occupational stress. Thus, our hypothesis was rejected. However, with regards to additional exposures, it was found that COVID-19 contamination was associated with pattern A, and shift work with pattern B.

With regards to the changes identified in the participants’ occupational characteristics, they reflect alterations in the structure and organization of work in hospitals. These are due to the increase in hospital admissions, on account of COVID-19, which has imposed on workers a work environment which is even more harmful to their health [5]. In addition, the high percentage of contamination of workers by COVID-19 (67.40%) is consistent with another reality of which these professionals had minimal experience. This is related to increased stress in patient care, the feeling of high risk in work performance, and concern for their own health [10].

Other studies have also demonstrated the effects of the pandemic on hospital workers’ health. According to Zhou et al. [5], symptoms of depression, anxiety, insomnia, and somatization are more severe in health teams than in the general population. There is also an increase in the level of occupational stress: Arafa et al. [37], when studying hospital workers in Egypt and Saudi Arabia, found that 55.9% had work stress, with 36.6% experiencing mild to moderate and 19.3% high to very high stress.

Due to this increasingly worrying scenario, another important factor to be evaluated is changes to these workers’ lifestyles, especially with regards to food, since they may result from occupational stress and may increase the risk of developing chronic, non-communicable diseases [38,39,40].

It is known that the relationship between food and stress occurs due to the considerable overlap of the physiological systems involved with food consumption and response to stress [41]. Due to this close relationship, stress can be associated with both an increase and decrease in food consumption [41,42]. At least temporarily, stress can also lead to other biological and behavioural changes, such as slower gastric emptying and increased preference for foods high in sugars and fats as a tool to manage temperament, tension, and stress [41].

Thus, investigation into food, through dietary patterns, is relevant, especially when considering changes in occupational factors imposed by the COVID-19 pandemic. In this study, three dietary patterns were identified before and during the pandemic, which accounted for 45.51 and 44.47% of the total sample variance, respectively.

Initially, pattern A was related to the bread and cereals, fruits, legumes, meat, milk and dairy products, fats, pasta and other preparations, and sweets groups, reflecting the traditional diet of Brazilian people, which is mostly composed of rice, beans, and meat of some kind [43]. During the pandemic, the food groups related to this pattern were tubers, fruits, vegetables, oilseeds, and eggs, indicating improvements in the quality of their diet, since these are considered indicators of healthy eating [24,44]. These changes were not associated with alterations in occupational stress levels; however, they were associated with the contamination of workers by COVID-19. Steele et al. [44], in their cohort study with 10,116 Brazilian adults from all regions of the country, found similar results; that is, a significant increase in the consumption of vegetables, fruits, and legumes during the pandemic.

These authors also explain that the COVID-19 pandemic may influence food in two ways: harmful or beneficial. The beneficial aspect, as seen in this study, refers to the improvement in diet through increased consumption of healthy indicators, which may arise from a possible concern by individuals to consume healthy foods, as an alternative to strengthening the immune system and the body’s defence against the coronavirus [44].

With regards to pattern B, the associated food groups before the pandemic were tubers, fruits, vegetables, oilseeds, and eggs; while during the pandemic, this was bread and cereals, meat, fats, pasta and other preparations, sweets, and sugary drinks. Unlike pattern A, changes in pattern B indicate a deterioration in the quality of food during the pandemic, indicated by the presence of unhealthy indicators, i.e., a source of sugars and fats [24]. However, these changes were not related to changes in occupational stress levels but to an increase in shift work.

The association between the change in pattern B and the increase in shift work verified in this study is in line with the literature, which indicates that shift work, defined as non-day, irregular, and/or rotational work, is associated with changes in workers’ lifestyles and, therefore, to future health problems [45,46]. Among these changes are those in eating behaviour, such as difficulty in maintaining a healthy diet, and/or increased consumption of high-calorie foods, rich in sugars and fats [45,47].

Farías et al. [48], in their study with health professionals from a hospital in Santiago, Chile, obtained similar results: lower diet quality index and vegetable consumption score, as well as lower frequency of meals, and higher omission of main meals. Furthermore, according to the systematic review carried out by Souza et al. [49], shift workers tend to present changes in meal patterns, skipping meals, and consuming food at unconventional times, increasing the consumption of unhealthy foods, especially those rich in saturated fats, and intake of sugary drinks.

In view of this, changes in the behaviour and dietary patterns of hospital workers, especially those related to pattern B, are a matter of concern, representing a serious risk to the health of these individuals, especially in the current context of a pandemic. The findings of this study, and others in the literature, indicate the need to establish strategies for better organization of routines and work in hospitals, to minimize the impacts of shift work and occupational stress, and to provide greater flexibility for workers to carry out their daily activities.

## 5. Conclusions

The present study aimed to verify the level of occupational stress, as well as the presence of workplace stressors, their changes, and associations with the dietary pattern of hospital workers, over time and comparing two points in time, before and during the COVID-19 pandemic.

Hence, it was possible to obtain a better and more comprehensive understanding of the work, food, and health conditions of workers in the hospital environment, which was provided in the innovative design of this work, effectively reflecting the impacts of the pandemic.

### 5.1. Research Limitations

The main limitations of this study refer to convenience sampling and self-reporting of contamination by COVID-19 by hospital workers. The first limitation is justified by the fact that the study was carried out during the pandemic, which made it difficult to conduct face-to-face interviews, due to high work demands, the turnover of professionals, and compliance with safety protocols. Despite this, the originality and innovative nature of this study are highlighted, comparing information before and during the pandemic and reflecting the changes imposed by the context of the pandemic. With regards to the self-reporting of contamination by COVID-19, it is believed that the impact of this measure on the results of this study may be minimized. The sample is composed of hospital workers, who by nature and workplace are assumed to hold greater and more accurate information about their health status and diagnosis of the disease than the general population.

### 5.2. Future Research

The COVID-19 pandemic has significantly changed the functional and lifestyle characteristics of the workers studied, resulting in an increase in the levels of stress and occupational stressors, as well as changes in these individuals’ dietary patterns, especially with regards to patterns A and B.

These findings are important for broadening the discussion regarding the health surveillance of these individuals in this current health crisis, both individually and collectively. They also constitute an important source of information to formulate corrective and preventive measures appropriate to the reality of hospital workers, with the objective of including not only healthy eating and living habits into their routines, but also non-invasive interventions related to stress at work, minimizing and/or preventing the risk of harm in later stages of life.

Finally, given the importance of professionals in the hospital sector in coping with the pandemic, further studies must be carried out to assess the post-pandemic context, that is, the implications left by the pandemic on their occupational, social, and health conditions, with emphasis on longitudinal assessments and representative samples.

## Figures and Tables

**Table 1 ijerph-20-04606-t001:** Descriptive analysis of the workers characteristics at baseline, Santo Antônio de Jesus city, 2019.

Characteristics	% (n)	Total (n)
Age (years)—Mean (SD)	32.60 (8.30)	218
Sex Women Men	75.20 (164) 24.80 (54)	218
Skin colour White Brown Black Other	12.80 (28) 48.60 (106) 35.30 (77) 3.30 (7)	218
Marital status Single Married/common law partner Divorced/separated Widowed	42.20 (92) 52.30 (114) 5.00 (11) 0.50 (1)	218
Educational level ≤High school ≥College	54.60 (119) 45.40 (99)	218
Income <3 minimum wages 3–5 minimum wages >5 minimum wages	12.40 (27) 64.20 (140) 23.40 (51)	218
Occupation Health professional Other	41.70 (91) 58.3 (127)	218
Smoking status Current smoker Ex-smoker Non-smoker	1.40 (3) 2.30 (5) 96.30 (210)	218
Alcohol consumption Yes No	51.80 (113) 48.20 (105)	218
Physical activity level Low Medium High	39.00 (85) 44.00 (96) 17.00 (37)	218
Family history of chronic diseases Yes No	87.20 (190) 12.80 (28)	218
Health self-perception Excellent/good Regular/bad	60.60 (132) 39.40 (86)	218
BMI classification Underweight/normal range Overweight/obese	51.80 (113) 48.20 (105)	218
WC classification Low risk Increased/high risk	46.80 (102) 53.20 (116)	218

SD: standard deviation/BMI: body mass index/WC: waist circumference.

**Table 2 ijerph-20-04606-t002:** Food group factor loadings of the workers’ dietary patterns identified for before and during the COVID-19 pandemic. Santo Antônio de Jesus city, 2019–2020.

Groups	Before the COVID-19 Pandemic			During the COVID-19 Pandemic		
	Pattern A	Pattern B	Pattern C	Pattern A	Pattern B	Pattern C
Breads and cereals	0.577	*	0.475	*	0.334	0.650
Tubers	*	0.555	*	0.619	*	*
Fruits	0.304	0.658	−0.342	0.680	*	*
Vegetables	*	0.667	*	0.744	*	*
Legumes	0.376	*	0.623	*	*	*
Oilseeds	*	0.660	*	0.650	*	*
Eggs	*	0.743	*	0.519	*	*
Meat	0.617	*	*	*	0.607	*
Milk and dairy products	0.330	*	*	*	*	0.655
Fats	0.583	*	*	*	0.411	0.643
Pasta and other preparations	0.695	*	*	*	0.766	*
Candy	0.663	*	*	*	0.390	*
Beverages	*	*	0.612	*	−0.325	0.502
Sugary drinks	*	*	−0.593	*	0.654	*
Kaiser–Meyer–Olkin’s test	0.709			0.713		

* The food group was not in the studied pattern.

**Table 3 ijerph-20-04606-t003:** Descriptive analysis of the workers’ sociodemographic, lifestyle, and health characteristics, and their associations with the dietary patterns over time. Santo Antônio de Jesus city, 2019–2020.

Characteristics	Pattern A			Pattern B			Pattern C		
	<P50	≥P50	*p* *	<P50	≥P50	*p* *	<P50	≥P50	*p* *
**Continuous form Mean (SD)**									
Age (years)	32.88 (8.14)	33.29 (8.52)	0.605	33.62 (8.79)	32.55 (7.82)	0.179	32.37 (7.97)	33.79 (8.63)	0.075
**Categorical form (% (n)**									
Sex Women Men	45.80 (80) 54.60(29)	51.50 (84) 45.40 (25)	0.318	51.50 (84) 45.40 (25)	48.50 (80) 54.60 (29)	0.318	52.70 (86) 41.70 (23)	47.30 (78) 58.30 (31)	0.059
Educational level ≤High school ≥College	49.20 (59) 51.00 (50)	50.80 (61) 49.00 (48)	0.773	55.80 (67) 42.90 (42)	44.20 (53) 57.10 (56)	0.009	41.30 (49) 60.70 (60)	58.80 (70) 39.30 (39)	<0.001
Income <3 minimum wages 3–5 minimum wages >5 minimum wages	49.10 (13) 46.30 (65) 60.8 (31)	50.90 (14) 53.70 (75) 39.20 (20)	0.042	39.60 (11) 53.70 (75) 46.10 (23)	60.40 (16) 46.30 (65) 533.90(28)	0.129	69.80 (19) 47.30 (66) 47.10 (24)	30.20 (8) 52.70 (74) 52.90 (27)	0.009
Occupation Health professional Other	51.10 (46) 49.20 (62)	48.90 (45) 50.80 (65)	0.771	51.60 (47) 48.80 (62)	48.40 (44) 51.20 (65)	0.627	50.50 (46) 50.50 (64)	49.50 (45) 49.50 (63)	1.000
Smoking status Current/ex-smoker Non-smoker	37.50 (3) 50.50 (106)	62.50 (5) 49.50 (104)	0.577	50.00 (4) 49.80 (104)	50.00 (4) 50.20 (106)	0.922	50.00 (4) 50.00 (109)	50.00 (4) 50.00 (109)	1.000
Alcohol consumption Yes No	50.00 (61) 50.00 (48)	50.00 (61) 50.00 (48)	1.000	48.00 (58) 52.60 (51)	52.00 (64) 47.40 (45)	0.385	53.30 (65) 45.80 (44)	46.70 (57) 52.40 (52)	0.148
Physical activity level Low Medium High	49.70 (36) 46.70 (52) 62.5 (20)	50.30 (37) 53.30 (60) 37.50 (12)	0.082	49.70 (36) 52.90 (60) 40.60 (13)	50.30 (37) 47.10 (53) 59.40 (19)	0.225	48.30 (35) 48.00 (54) 60.90 (20)	51.70 (38) 52.00 (59) 39.10 (12)	0.177
Family history of chronic diseases Yes No	50.00 (95) 50.00 (14)	50.00 (95) 50.00 (14)	1.000	51.10 (97) 42.90 (12)	48.90 (93) 57.10 (16)	0.316	50.30 (96) 48.20 (13)	49.70 (94) 51.80 (15)	0.886
Health self-perception Excellent/good Regular/bad	53.00 (70) 45.3 (39)	47.00 (62) 54.70 (47)	0.268	47.70 (63) 53.50 (46)	52.30 (69) 46.50 (40)	0.406	50.00 (66) 48.80 (42)	50.00 (66) 51.20 (44)	0.867
BMI classification Underweight/normal range Overweight/obese	50.50 (55) 49.50 (54)	49.50 (53) 50.50 (56)	0.924	48.10 (52) 51.80 (57)	51.90 (56) 48.20 (53)	0.503	49.10 (53) 50.90 (56)	50.90 (55) 49.10 (54)	0.774
WC classification Low risk Increased/high risk	55.10 (52) 46.20 (57)	44.90 (42) 53.80 (67)	0.081	47.10 (44) 52.20 (65)	52.90 (50) 47.80 (59)	0.333	51.30 (48) 49.00 (61)	47.80 (46) 51.00 (64)	0.669

SD: standard deviation/BMI: body mass index/WC: waist circumference. * Student’s *t* and Pearson’s chi-squared test.

**Table 4 ijerph-20-04606-t004:** Generalized Estimating Equations models for the relationship between occupational stress and work stressors and the dietary pattens over time. Santo Antônio de Jesus city, 2019–2020.

Characteristics	Dietary Patterns		
	Pattern A Coefficient (95%CI) *p* *	Pattern B Coefficient (95%CI) *p* *	Pattern C Coefficient (95%CI) *p* *
Occupational stress QIC No Yes	305940 Reference 0.26 (−0.51; 1.03) 0.51	304316 Reference −0.79 (−0.78; 0.20) 0.12	304084 Reference −0.22 (−0.99; 0.54) 0.57
Shift work QIC No Yes	304651 Reference 0.41 (−0.21; 1.02) 0.19	302.894 Reference 0.65 (0.05; 1.25) 0.03	305073 Reference 0.28 (−0.31; 0.87) 0.35
Weekly workload QIC <44 h >44 h	305711 Reference 0.16 (−0.50; 0.81) 0.63	304297 Reference −0.15 (−0.86; 0.56) 0.67	304288 Reference 0.20 (−0.46; 0.86) 0.55
COVID-19 infection QIC No Yes	303.677 Reference 0.68 (0.08; 1.29) 0.03	305193 Reference 0.06 (−0.55; 0.66) 0.86	301845 Reference 0.13 (−0.54; 0.80) 0.70

Sample size: 218. CI: confidence interval. * Generalized Estimating Equation models—GEE.

## Data Availability

Not applicable.

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
