# Peer review of "The Effects of Workplace Stressors on Dietary Patterns among Workers at a Private Hospital in Recôncavo of Bahia, Brazil: A Longitudinal Study before and during the COVID-19 Pandemic"

_ijerph, 2023, doi:10.3390/ijerph20054606_

Round 1

Reviewer 1 Report

Workplace stressors are associated with changes in dietary patterns of hospital workers in the Recôncavo of Bahia: a study before and during the COVID-19 pandemic.

Dear Authors,

It is possible to consider your paper entitled "Workplace stressors are associated with changes in dietary patterns of hospital workers in the Recôncavo of Bahia: a study before and during the COVID-19 pandemic" for publication only after some revision. I have listed my comments below.

1.     The title of the manuscript indicates the sample included all hospitals in Recôncavo of Bahia, while in real data collected from one hospital only. It would also be beneficial to rephrase the title of the study.

2.     The abstract is not well organized. The way of expressing ideas is not clear. Authors need to rewrite it according to the followings sequence

*      Introduction: should be brief and attractive to the reader at the same time

*      Research significance: This usually answers the question: Why did you do this research?

*      Methodology.

*      Results.

*      Originality of the research and practical implications.

3.     The topic is directly addressed in the introduction. The problem statement is clear and concise. Research problems formulated in a scientifically sound manner. There are, however, a few amendments that should be made by the authors in this section.

*     There is a lack of clarity regarding the main variables of the study in the introduction section, which needs to be addressed by the authors.

*     I recommend adding introductory paragraph about working environment and the increase in psychological distress and the occupational stress levels for individuals before and after covid-19 in general in other settings (such as; Mineworkers and  Geological Investigators) to highlight on the importance of the topic universally, After that, you can discuss and address the health setting. I recommend adding these two newly published articles and pls cite more settings with updated articles.  

1.     Alrawad, M., Lutfi, A., Alyatama, S., Elshaer, I. A., & Almaiah, M. A. (2022). Perception of occupational and environmental risks and hazards among mineworkers: a psychometric paradigm approach. International journal of environmental research and public health19(6), 3371.

2.     Tao, S., Hao, J., & Yu, J. (2022). How Does Perceived Organizational Support Reduce the Effect of Working Environmental Risk on Occupational Strain? A Study of Chinese Geological Investigators. International Journal of Environmental Research and Public Health20(1), 51.

*     The references used in the introduction are somewhat outdated. Authors must update it with recently published papers to provide a sound scientific foundation for the work. There is only one reference in 2020, the rest are obsolete.

*     Although the research problem is well-written, the research gap is not clearly defined. It is imperative for the authors to clarify how their research contributes to the filling of a theoretical, conceptual, or methodological gap.  Researchers should include a literature review section in their manuscript in order to identify gaps in their research and clarify them. The content of this section should be extensive. Recent publications must be cited by the authors.

4.     The research goals and the importance of the research are specific, clear, and straightforward objectives underlying the research that have been formulated in a straightforward manner. In the field of specialization, research plays an imperative role, and its application is advantageous.

5.     The literature review section must be added.  A number of relevant studies should be incorporated.  Also, the results of the previous studies must be incorporated and drawn up in order to provide more information.

6.     Research methodology takes into account the study's objectives. In accordance with scientific standards and guidelines, systematic methods have been applied. However, authors must clarify more about the reason for choosing one hospital only in this section.

7.     The results of the study are presented in an organized and accurate manner. The study's findings are based on sound scientific principles. It was found that the study's results were in agreement with those found in prior research and theories. Although the authors cite recent publications, older publications may be limited. Particularly those that date back to 2017 and before.

8.      In the discussion section, in the last paragraph (lines 410-420), which discusses research limitations, it would be better to move to the conclusion in the subheading titled Research limitations and future research.

9.     In light of the research objectives and its novelty, the study's conclusion reflects both. However, the last paragraph in the discussion section (line 410-420) should be moved to the conclusion section under the subheading titled limitations of the research and future research. Additionally, the following points need to be added and clarified.

A.    The study practical and theoretical implications need more clarifications. Authors must add them in the conclusion section.

B.    What are the future studies streams based on this study limitations. Must be added. 

Finally, I recommend that this paper be accepted with major revision.

Author Response

Dear Editor and Referees,

Thank you so much for considering our manuscript and giving us this opportunity to improve our work. We carefully considered all your comments and suggestions, as following attached.

Sincerely,

Lorene Gonçalves Coelho.

Reviewer 2 Report

This is a well done study, but needs a minor revision. More relevant studies should be cited in the introduction section.

Author Response

Dear Editor and Referees,

Thank you so much for considering our manuscript and giving us this opportunity to improve our work. We carefully considered all your comments and suggestions, as following.

Referee 2

Does the introduction provide sufficient background and include all relevant references? Can be improved - We appreciate this thought. We improved the introduction and updated the references.

We hope to have satisfactorily clarified all the issues pointed and remain available for any further considerations or suggestions.

Sincerely,

Lorene Gonçalves Coelho.

Reviewer 3 Report

Workplace stressors are associated with changes in dietary patterns of hospital workers before and during the COVID-19 pandemic is an interesting research. 

The main limitations of this study refer to convenience sampling and self-reporting of contamination by COVID-19 by hospital workers. The first limitation is justified by the fact that the study was carried out during the pandemic, which made it difficult to conduct face-to-face interviews, due to high work demands, the turnover of professionals, and compliance with safety protocols. 

Authors conclude that the COVID-19 pandemic has significantly changed the functional and lifestyle characteristics of the workers studied, resulting in an increase in the levels of stress and occupational stressors, as well as changes in these individuals' dietary patterns. 

It would be interesting to research what was happened after the intensive pandemic activities of health professionals, and is there any consequences in nutrition behaviours. 

Suggestions: could me make any link in nutritional habits  with mental health/conditions?

References could be improved. 

Author Response

Dear Editor and Referees,

Thank you so much for considering our manuscript and giving us this opportunity to improve our work. We carefully considered all your comments and suggestions, as following.

Referee 3

It would be interesting to research what was happened after the intensive pandemic activities of health professionals and is there any consequences in nutrition behaviours - We appreciate this thought. We have plans to come back to the hospital to do a new and final evaluation.

Suggestions: could me make any link in nutritional habits with mental health/ conditions? - Thank you for this suggestion. Unfortunately, we do not have mental health data in our study.

References could be improved - We appreciate this thought. We improved it.

We hope to have satisfactorily clarified all the issues pointed and remain available for any further considerations or suggestions.

Sincerely,

Lorene Gonçalves Coelho.

Round 2

Reviewer 1 Report

Workplace stressors are associated with changes in dietary patterns of hospital workers in the Recôncavo of Bahia: a study before and during the COVID-19 pandemic.

Dear Authors,

Please accept my sincere thanks for your clear and relevant responses, and for the changes you have made to the manuscript. Several issues have been clarified as a result of these discussions; however, some minor issues may still require improvement.

1.     As far as I am concerned, the way the title is constructed is inappropriate.I suggest the following title “The effects of workplace stressors on dietary patterns among workers at a private hospital in Recôncavo, Bahia: a longitudinal study before and during the COVID-19 pandemic.”

2.     I would suggest that you include one or two paragraphs before sections 5.1 and 5.2 that summarize the research objectives and the novelty of your study.

A third round of revisions is not necessary in my opinion. The editor can handle corrections.  I suggest that this paper be accepted.

Author Response

Dear Editor and Referees,

Thank you so much for considering our manuscript and giving us the opportunity to improve our work once again. We carefully considered all your comments and suggestions, as following.

  1. As far as I am concerned, the way the title is constructed is inappropriate. I suggest the following title “The effects of workplace stressors on dietary patterns among workers at a private hospital in Recôncavo, Bahia: a longitudinal study before and during the COVID-19 pandemic.”

Thank you for these thoughts. We improved the title by considering the Referee’s suggestion.

  1. I would suggest that you include one or two paragraphs before sections 5.1 and 5.2 that summarize the research objectives and the novelty of your study.

We appreciate this consideration. We improved the conclusion by considering the Referee’s suggestion.

We hope to have satisfactorily clarified all the issues pointed and remain available for any further considerations or suggestions.

Sincerely,

Lorene Gonçalves Coelho.